# CAMILA: Context-Aware Masking for Image Editing with Language Alignment

**Hyunseung Kim**[1,2*]  **Chiho Choi**[1]  **Srikanth Malla**[1]  **Sai Prahladh Padmanabhan**[1]
**Saurabh Bagchi**[2]  **Joon Hee Choi**[1]
[1]Samsung Semiconductor, USA   [2]Purdue University
{kim4061, sbagchi}@purdue.edu
{chiho1.choi, srikanth.m, sai.prahladh, jh4.choi}@samsung.com

## Abstract

Text-guided image editing has been allowing users to transform and synthesize images through natural language instructions, offering considerable flexibility. However, most existing image editing models naively attempt to follow all user instructions, even if those instructions are inherently infeasible or contradictory, often resulting in nonsensical output. To address these challenges, we propose a context-aware method for image editing named as CAMILA (**C**ontext-**A**ware **M**asking for **I**mage Editing with **L**anguage **A**lignment). CAMILA is designed to validate the contextual coherence between instructions and the image, ensuring that only relevant edits are applied to the designated regions while ignoring non-executable instructions. For comprehensive evaluation of this new method, we constructed datasets for both single- and multi-instruction image editing, incorporating the presence of infeasible requests. Our method achieves better performance and higher semantic alignment than state-of-the-art models, demonstrating its effectiveness in handling complex instruction challenges while preserving image integrity.

## 1 Introduction

In recent years, the growing demand for visual content has made image editing essential across various fields. With advancements in technology, text-guided image editing has emerged as a powerful tool, enabling users to manipulate images using natural language instructions [4, 16, 10, 42, 11, 12]. This innovation has streamlined the editing process, enabling users to perform sophisticated edits. Among these advancements, diffusion-based models have particularly excelled in image generation [13, 38, 39, 30, 48, 45, 3] and editing tasks [18, 8, 12, 4, 42, 48]. However, models relying on simple text encoders such as CLIP [35] struggle to achieve user-intended fine-grained edits. These difficulties become more apparent when the editing prompt involves multi-step instructions with intricate details.

To address this limitation, recent research has introduced two notable improvements in model design. First, the CLIP-like text encoder has been replaced by Multimodal Large Language Models (MLLMs) [16, 10]. These models effectively parse user instructions and interpret textual prompts, improving the capabilities of natural language understanding. Second, regions requiring editing within the image are identified and modified using various methods, such as cross-attention maps and segmentation models, to align each edit prompt with its corresponding regions [11, 25]. Although the region-based image editing model [11] shows more effective results on multi-instruction tasks than other state-of-the-art methods, its attention maps often fail to consistently align with intended editing regions. This misalignment is especially pronounced when modifications involve spatial relationships or regions not directly associated with primary instruction keywords.

---

*Work done during an internship at Samsung Semiconductor, USA.

39th Conference on Neural Information Processing Systems (NeurIPS 2025).

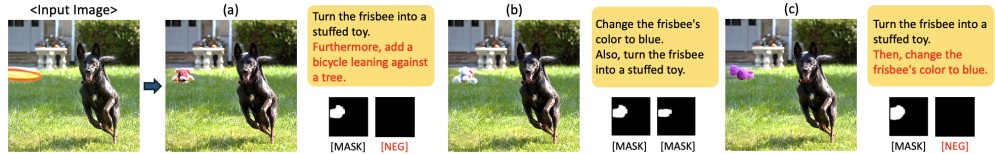

Figure 1: Three scenarios demonstrate how our method handles context-aware multi-instruction editing across various combinations of feasible and infeasible prompts. By leveraging [MASK] and [NEG] specialized tokens, it accurately identifies executable instructions.

These limitations become evident in multi-instruction scenarios containing challenging instructions that cannot be directly applied to the current image. Such instructions may request alterations to non-existent objects, logically inconsistent modifications, or edits that are incompatible with the image's content. Parsing and interpreting such inputs makes editing systems impractical, introducing suboptimal edits or even unrealistic, incoherent images. Additionally, relying on pretrained Large Language Models [32] to parse or reorganize these instructions introduces further complexity in the editing pipeline and increases the potential for errors at intermediate steps. Any misinterpretation or bias in LLM output may propagate downstream, leading to incorrect region selection or over-editing.

Despite the growing research interest in comprehensive image editing, most existing methods overlook instruction executability, often leading to over-edited results. Our proposed approach addresses these concerns by explicitly assessing the executability of the instruction throughout the editing process. Building on pioneering research in this domain, we leverage the MLLM to jointly interpret both text instructions and images, then we extend its capabilities to enable image editing with context awareness. Here, *context* refers to the model's ability to interpret the relevance of various instructions within a given image, allowing it to focus on applicable regions while ignoring irrelevant areas. A key feature of CAMILA is the use of specialized tokens and broadcast mechanism. Our model assigns [MASK] tokens to editable regions and [NEG] tokens to suppress irrelevant edits. The following broadcasting module then consistently aligns token assignments with user prompts. Overall, our context-aware pipeline helps to validate the coherence of instructions, resulting in improved performance across all image editing scenarios, including non-executable prompts.

To properly evaluate our approach, we extend the conventional single- and multi-instruction image editing tasks by introducing the possibility of non-executable prompts. This results in new evaluation scenarios: *Context-Aware Image Editing* that evaluate how the model handles the number of instructions and the presence of infeasible requests within the same sequence. We compare our method against several state-of-the-art baselines, observing substantial improvements in editing accuracy, particularly L1 and L2 distances, as well as enhanced performance on CLIP and DINO scores, with a human preference-based evaluation also indicating strong performance.

Our main contributions to this work are as follows:

- We introduce a context-aware image editing model that precisely identifies prompt executability and corresponding editing regions, allowing user-aligned and consistent modifications.
- We propose a new task setting: *Context-Aware Image Editing*. New datasets are created to evaluate model behavior and context-awareness in challenging scenarios.
- Our model demonstrates significant improvements over existing methods in varying evaluation scenarios, achieving lower pixel-level errors and higher semantic alignment, while also showing qualitative superiority in effectively handling complex instructions.

Note that we formally define 'non-executable instruction' as any request that cannot be executed given the visual constraints or inherent semantics of the image. Our source code is available at https://github.com/hk-repo/CAMILA.

## 2 Related Works

**Multimodal Large Language Models.** Multimodal Large Language Models (MLLMs) [24, 27, 9, 41, 51, 26] integrate multiple modalities, such as images and text. Recent MLLMs have advanced to handle complex tasks such as referring visual grounding [50, 23, 7, 43], which aims to distinguish specific objects based on context. Additionally, MLLMs have been applied to image editing task [16,

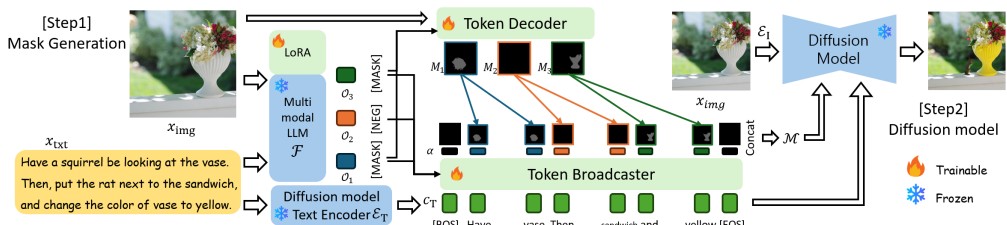

Figure 2: The architecture of CAMILA begins by jointly processing the image $x_{\text{img}}$ and text instructions $x_{\text{txt}}$ using an MLLM. Output tokens are classified as either [MASK] or [NEG], indicating regions to modify or leave unchanged. These tokens are aligned with the text embeddings using the Token Broadcaster, and the final binary mask is generated by the Token Decoder. The mask is then applied in a diffusion model to produce the edited image.

10]. For instance, SmartEdit [16] improves instruction comprehension with bidirectional interactions between image and text, while MGIE [10] jointly trains an MLLM and diffusion model to guide editing tasks with visual-aware instructions. However, these models often lack context-awareness and fail to distinguish between relevant and irrelevant prompts. We thus break new ground by being the first to incorporate a context-aware MLLM specially for image editing. Unlike prior research, we do not limit our scope to single instruction tasks, enabling our model to handle both multi and context-aware instructions.

**Image Editing by Diffusion Model.** Diffusion models have become prominent in image editing [37, 1, 30, 29, 18, 8, 12, 42, 48, 49]. While text-guided image editing enable basic modifications, instruction-based image editing offers more nuanced control by interpreting complex, user-directed commands via natural language. InstructPix2Pix [4] introduced a dataset combining GPT-3-generated texts [5] and Prompt2Prompt-based images [12], which powers natural language-guided editing. MGIE [10] utilizes an MLLM with visual-aware instructions for editing, and FoI [11] uses cross-attention maps for multi-instruction scenarios. However, these methods struggle with ambiguous or incorrect instructions, as they lack mechanisms to interpret prompt feasibility. This limitation often leads to unintended modifications when the model encounters unclear instructions.

## 3 Preliminary

We briefly introduce InstructPix2Pix (IP2P) [4], a standard framework for instruction-guided image editing and its cross-attention mechanism. This overview serves as the background for our work.

### 3.1 InstructPix2Pix

IP2P [4] is built upon Stable Diffusion [38] to modify images based on textual instructions. In this framework, conditioning on both input image and text instructions is necessary for guiding diffusion network to produce editing results aligned with user instruction. The input image $x_{\text{img}}$ is first encoded into a latent vector $z$ by the encoder $\mathcal{E}_{\text{I}}$. At each time step $t$, the noisy latent vector $z_t$ is progressively denoised by the score network. Then, the denoised latent vector $z$ is decoded into the output image.

To achieve conditional generation, diffusion models often employ classifier-free guidance [14], which eliminates the need for an external classifier. In their score network, two conditioning factors are introduced for use during inference: the image conditioning $c_I$ and the text instruction conditioning $c_T$. $c_I$ and $c_T$ are the encoded outputs from the image encoder $\mathcal{E}_{\text{I}}$ and the text encoder $\mathcal{E}_{\text{T}}$, respectively. The final score estimation $\tilde{e}_\theta(z_t, c_I, c_T)$ is computed as follows:

$$\begin{aligned} \tilde{e}_\theta(z_t, c_I, c_T) = e_\theta(z_t, \varnothing, \varnothing) &+ s_I \cdot (e_\theta(z_t, c_I, \varnothing) - e_\theta(z_t, \varnothing, \varnothing)) \\ &+ s_T \cdot (e_\theta(z_t, c_I, c_T) - e_\theta(z_t, c_I, \varnothing)). \end{aligned} \tag{1}$$

In this equation, $e_\theta(z_t, \varnothing, \varnothing)$ represents the base score prediction without any conditioning applied. The second term modulates the score with image conditioning $c_I$, where $s_I$ modulates how much the model preserves the characteristics of the input image. Similarly, the last term incorporates text conditioning $c_T$, with $s_T$ controls the degree of adherence to the edit instruction provided.

## 3.2 Cross Attention in Stable Diffusion

IP2P employs cross-attention network modulation within the denoising U-Net architecture of the Stable Diffusion network. A key component is the cross-attention layer, which generates attention maps $\mathcal{A} \in \mathbb{R}^{r \times r \times m}$, where $r$ is the spatial size and $m$ is the number of text tokens. Several studies [2, 6, 11] have shown that cross-attention maps with $r = 16$ capture the most significant semantic information, compared to maps at other spatial resolutions. Thus, by modulating the computation of these cross-attention layers, it is possible to alter the image, as adjustments in the attention maps guide the model's focus on specific aspects of the text and image content [8, 44].

## 4 Methods

We build our framework upon a pretrained MLLM [27] and diffusion model [38], but our key contribution lies in explicitly assessing the executability of instructions and leveraging specialized tokens to guide editing process in diffusion model. A key feature of our approach is its ability to validate the contextual coherence between instructions and the image, ensuring that only relevant edits are applied to designated regions while ignoring non-executable instructions. This context-aware mechanism distinguishes our method from existing MLLM-based approaches [10, 16], establishing executability filtering and context-awareness as new modeling objectives for MLLM-based image editing.

### 4.1 Architecture

The architecture of CAMILA is shown in Figure 2. Given an image $x_{\text{img}}$ and text instructions $x_{\text{txt}}$, both inputs are jointly processed by the MLLM $\mathcal{F}$. The model is designed to encode and combine the visual and textual inputs, enabling it to capture the relationships between the textual instructions and corresponding regions in the image. Specifically, the image is processed through a vision encoder, while the text instructions are tokenized and processed by a language encoder. These representations are then combined into a unified sequence within the MLLM architecture, which interprets the joint context of the image and instructions. The output sequence $\mathcal{O}$ is generated from the image input $x_{\text{img}}$ and text input $x_{\text{txt}}$. Each output token $\mathcal{O}_i$ in $\mathcal{O} = \{\mathcal{O}_1, \mathcal{O}_2, \ldots, \mathcal{O}_n\}$, where $n$ denotes the number of generated tokens, is classified as either a [MASK] or [NEG] token. The [MASK] tokens correspond to regions of the image that are to be modified based on the text instructions, while the [NEG] tokens indicate areas of the image that should remain unaffected.

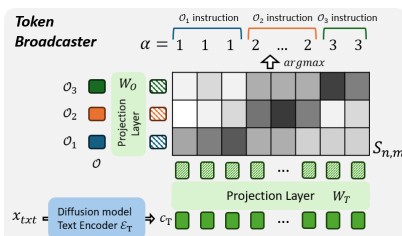

Figure 3: **Architecture of the Token Broadcaster.** It calculates similarity between MLLM output tokens and encoded text features, assigning each output token to the text embedding that best matches its corresponding semantic region.

By combining the visual and textual inputs, the MLLM is able to determine the relevance of each instruction to specific regions in the image, ensuring that only applicable edits are applied. This joint processing aligns each generated output token, either [MASK] or [NEG], with specific instructions. The [MASK] tokens are decoded, resulting in masks that accurately highlight the regions in the image that require modification according to the instructions. This targeted approach improves the precision of the editing process by ensuring that modifications are applied solely to relevant areas. The following content will elaborate on how [MASK] and [NEG] tokens are aligned with instructions by Token Broadcaster and how [MASK] tokens are decoded into the actual editing mask by Token Decoder.

### 4.2 Token Broadcaster and Token Decoder

**Token Broadcaster.** The output sequence $\mathcal{O}$ generated by the MLLM is processed by the Token Broadcaster module to ensure that the [MASK] and [NEG] tokens align accurately with the corresponding text embeddings. As illustrated in Figure 3, the text instructions $x_{\text{txt}}$ are embedded through the text encoder $\mathcal{E}_{\text{T}}$ of the diffusion model, resulting in a set of text embeddings $c_T$. Using the diffusion model's text encoder $\mathcal{E}_{\text{T}}$ allows the model to ensure that the generated editing masks will align precisely with $c_T$, facilitating integration into the diffusion model.

The MLLM output tokens $\mathcal{O}$ and the text embeddings $c_T$ reside in different latent spaces, so we need to align them into a single space. Many studies [20, 46] use cosine similarity-based alignment to measure and organize relationships or similarities between different modalities. We project them into a shared space for alignment by applying trainable transformations $W_O$ and $W_T$ to each, directly within the similarity matrix:

$$S_{i,j} = \frac{(\mathcal{O}_i W_O) \cdot (c_{Tj} W_T)}{\|(\mathcal{O}_i W_O)\| \|(c_{Tj} W_T)\|}, \tag{2}$$

where each element $S_{i,j}$ represents the cosine similarity score between the $i$-th transformed output token $(\mathcal{O}_i W_O)$ and the $j$-th transformed text embedding $(c_{Tj} W_T)$, indicating their compatibility in the shared latent space.

To convert similarity scores into alignment probabilities, a softmax is applied along each column of $S$. For each text embedding $j$, we then determine the index $\alpha_j$ that maximizes this probability:

$$\alpha_j = \arg\max_i \left( \frac{\exp(S_{i,j})}{\sum_k \exp(S_{k,j})} \right), \forall j \in \{1, 2, \ldots, m\}, \tag{3}$$

where $m$ denotes the length of text embeddings. This alignment process ensures that each text embedding maps to the output token best reflecting its semantic region within the image.

**Token Decoder.** The Token Decoder processes tokens differently based on their type: only tokens labeled as [MASK] are converted into editing masks, while [NEG] tokens are directly replaced with black masks, indicating regions where no modification is applied. Designed as a two-layer Transformer decoder, the Token Decoder generates a set of binary masks $M_1, M_2, \ldots, M_n$, each specifying regions of the image to be edited according to the text instructions.

In the first decoder layer, we employ a cross-attention mechanism between image and text embeddings. This allows the model to extract contextually relevant features from the image that are aligned with the text instructions. By attending to both modalities, the decoder effectively maps the semantic content of the text to corresponding regions in the image. The second decoder layer further refines this information by incorporating the [MASK] tokens into the key and value projections of the attention mechanism. This enables the model to focus more precisely on the regions identified by each [MASK] token. After the second decoder layer, these intermediate masks are passed through sigmoid thresholding to produce the final 0-1 binary masks, denoted as $M_i$. Through this process, Token Decoder is able to generate the final binary mask $M_i$, with each mask serving as an editing mask for the corresponding MLLM output tokens $\mathcal{O}_i$, defining the specific areas of the image to be modified.

### 4.3 Diffusion Model

For each text embedding $j$, the alignment index $\alpha_j$ determines the specific binary mask $M_{\alpha_j}$ to be used. The individual masks are concatenated to form a unified binary mask $\mathcal{M}$, which is then used in the diffusion model to guide the editing process:

$$\mathcal{M} = \text{concat}(M_{\alpha_1}, M_{\alpha_2}, \ldots, M_{\alpha_m}). \tag{4}$$

This binary mask $\mathcal{M}$ ensures that each region is modified according to alignment indices from the Token Broadcaster, enabling precise, context-aware edits that reflect the intended modifications.

We modulate the cross-attention layers of the diffusion model, focusing specifically on the 16-sized cross-attention map, which captures the most semantically relevant features, as explained in Section 3.2. The U-Net's cross-attention map $\mathcal{A}$ is modulated using the following equation:

$$\mathcal{A}' = \text{softmax} \left( \frac{\mathcal{X} \odot \mathcal{M} + \mathcal{Y} \odot (1 - \mathcal{M})}{\sqrt{d}} \right), \tag{5}$$

where $d$ is the latent projection dimension, $\mathcal{X} = Q_{I,T} K_{I,T}^{\mathsf{T}}$, and $\mathcal{Y} = Q_{I,\varnothing} K_{I,\varnothing}^{\mathsf{T}}$. In this formulation, $Q_{I,T}$ and $K_{I,T}$ represent the query and key projections in $e_\theta(z_t, c_I, c_T)$, respectively, while $Q_{I,\varnothing}$ and $K_{I,\varnothing}$ are the query and key projections in $e_\theta(z_t, c_I, \varnothing)$.

This modulation approach leverages $\mathcal{A}$ to align each text embedding precisely with the regions specified by the concatenated binary mask $\mathcal{M}$, enhancing editing accuracy by concentrating on the relevant areas as dictated by the instructions. Then, the binary mask $\mathcal{M}$ selectively applies the text-conditioned attention map $\mathcal{X}$ to editable regions and $\mathcal{Y}$ to unaltered areas, ensuring that only the specified areas are modified. By modulating the attention layer as in Equation (5), we generate the final output image following the score estimation formulated in Equation (1).

## 4.4 Training Details

**Training Loss Function.** The training of our MLLM-based approach is optimized with four primary loss components, each designed to target a specific aspect of model performance for accurate token classification, alignment, and mask generation. The total loss $\mathcal{L}_{\text{main}}$ is formulated as follows:

$$\mathcal{L}_{\text{main}} = \lambda_1 \mathcal{L}_{\text{CE}}^{\text{token}} + \lambda_2 \mathcal{L}_{\text{CE}}^{\text{broadcast}} + \lambda_3 \mathcal{L}_{\text{dice}} + \lambda_4 \mathcal{L}_{\text{BCE}}, \tag{6}$$

where $\lambda_1, \lambda_2, \lambda_3, \lambda_4$ are hyperparameters that balance the influence of each loss component.

The first element, token classification loss $\mathcal{L}_{\text{CE}}^{\text{token}}$, applies cross-entropy (CE) loss to the MLLM output tokens. The second element, broadcasting alignment loss $\mathcal{L}_{\text{CE}}^{\text{broadcast}}$, also utilizes CE loss to align MLLM output tokens with their respective text embeddings, ensuring precise correspondence between instructions and image regions. For mask quality, the mask dice loss $\mathcal{L}_{\text{dice}}$ measures overlap between predicted and ground truth masks, encouraging accurate spatial targeting. Lastly, the binary cross-entropy loss $\mathcal{L}_{\text{BCE}}$ enforces accuracy at the pixel level in the generated mask.

**Trainable Parameters.** To efficiently fine-tune the pre-trained MLLM while preserving its learned knowledge, we adopt the Low-Rank Adaptation technique [15]. In our training, we freeze the vision backbone and text encoder of the MLLM, while the remaining parts of the model are fine-tuned. Additionally, the Token Broadcaster and Token Decoder are also trained, ensuring that the model aligns the output tokens with the text instructions and generates accurate masks for the diffusion model. Training is more efficient since only the MLLM and lightweight modules are updated, unlike other methods that jointly fine-tune both MLLM and diffusion model. All other training details are provided in Section A.

## 4.5 Surrogate Module Training for Enhanced Masking

To further improve the quality of the binary mask $\mathcal{M}$ provided to the diffusion model, we conduct additional training beyond the initial MLLM training. Through empirical analysis, we found that certain outputs misalign with the description of the goal image. To better align the generated image with the intended modifications, we consider it useful to focus on improving CLIP-T score, which measures the similarity between the global description and the generated image. By optimizing the model for a higher CLIP-T score, we aim to generate higher quality binary masks, which lead to improved quality in the final output image.

However, due to the inherent complexity and the large number of steps involved in the forward pass of the diffusion model, directly backpropagating the loss from the final output image through the diffusion model to the MLLM is infeasible. To address this limitation, we develop a lightweight surrogate module that approximates the CLIP-T score based on the input image $x_{\text{img}}$, the edit instruction $x_{\text{txt}}$, and the binary masks $\mathcal{M}$. Designed as a single-layer transformer, the surrogate module offers a streamlined alternative to the complex, multi-step diffusion model. It is trained using a mean squared error (MSE) loss between the actual CLIP-T score and the predicted CLIP-T score. During this training phase, all other parts of the model are kept frozen, and only the surrogate module is updated. The overall loss function for training the surrogate module is formulated as:

$$\mathcal{L}_{\text{surrogate}} = \mathbb{E}\left[ \left( \text{CLIP-T}_{\text{output}} - \text{CLIP-T}_{\text{surrogate}} \right)^2 \right], \tag{7}$$

where $\text{CLIP-T}_{\text{output}}$ and $\text{CLIP-T}_{\text{surrogate}}$ denote the actual CLIP-T score of the target output and the predicted score, respectively. This approach ensures that the surrogate module learns to accurately estimate the CLIP-T score without requiring multi-step backpropagation of the diffusion model.

**Refining Mask Generation via Surrogate Module.** Once the surrogate module is fully trained, we use estimated values to fine-tune the MLLM, Token Broadcaster, and Token Decoder. In this stage, the surrogate module is kept frozen, and the focus is on improving mask generation to maximize the predicted CLIP-T score. The objective is to modify the MLLM's outputs to generate binary masks with a higher CLIP-T score when processed by the diffusion model.

During training, the loss function is augmented to include both $\mathcal{L}_{\text{main}}$ as well as the MSE loss between the predicted CLIP-T score and the oracle CLIP-T score. The updated loss $\mathcal{L}_{\text{updated}}$ is defined as:

$$\mathcal{L}_{\text{updated}} = \mathcal{L}_{\text{main}} + \lambda_5 \mathcal{L}_{\text{MSE}}, \tag{8}$$

where $\mathcal{L}_{\text{MSE}}$ is the MSE loss between the predicted and oracle CLIP-T score, and $\lambda_5$ is a hyperparameter controlling the weight of the CLIP-T score loss.

Table 1: **Quantitative comparison across multi-instruction and context-aware instruction tasks.** Our model demonstrates overall superior performance, especially excelling in the context-aware instruction task. This highlights our method's superb capability to handle context-aware instructions with high precision, applying edits that closely align with the intended modifications without over-editing. **Bold** and underlining indicates the best and the second-best performance for each metric.

| Method | Multi Instruction | | | | | Context-Aware Instruction | | | | |
|---|---|---|---|---|---|---|---|---|---|---|
| | L1↓ | L2↓ | CLIP-I↑ | DINO↑ | CLIP-T↑ | L1↓ | L2↓ | CLIP-I↑ | DINO↑ | CLIP-T↑ |
| IP2P [4] | 0.1402 | 0.0526 | 0.8327 | 0.7122 | 0.2977 | 0.1460 | 0.0514 | 0.7975 | 0.6429 | 0.2715 |
| MGIE [10] | 0.1639 | 0.0777 | 0.8205 | 0.6723 | 0.2787 | 0.1592 | 0.0750 | 0.8090 | 0.6519 | 0.2637 |
| SmartEdit [16] | 0.1295 | 0.0573 | 0.8630 | 0.7516 | 0.2971 | 0.1111 | 0.0495 | 0.8739 | 0.7726 | 0.2824 |
| FoI [11] | 0.1054 | 0.0385 | 0.8811 | 0.8096 | 0.2941 | 0.0891 | 0.0284 | 0.8895 | 0.8190 | 0.2888 |
| CAMILA (ours) | **0.0945** | **0.0366** | **0.8980** | **0.8392** | **0.2984** | **0.0661** | **0.0222** | **0.9296** | **0.8932** | **0.3006** |

## 5 Evaluation

### 5.1 Task Categorization

For a comprehensive assessment, we evaluate our method on both single-instruction tasks aligned with standard benchmarks, and multi-instruction image editing tasks that require multiple edit turns in a single sequence. In a single-instruction scenario, a single directive is tested either in a single-turn or multi-turn setting, whereas multi-instruction tasks involve multiple directives that must be applied simultaneously. We further divide multi-instruction tasks into two types: *Multi-instruction Image Editing*, which includes only applicable instructions, and *Context-Aware Instruction Image Editing*, which includes a mix of applicable and non-applicable instructions.

### 5.2 Evaluation Settings

**Datasets:** For evaluating single instruction tasks, we use the MagicBrush [47] dataset, which covers both single-turn and multi-turn scenarios as detailed in Section 6, along with the EMU [40] dataset. However, the literature lacks dedicated benchmark datasets for multi-instruction or context-aware instruction editing. To address this gap, we introduce two new tasks and curate corresponding datasets: *Multi-instruction Image Editing* and *Context-Aware Instruction Image Editing* as detailed in Section 5.3. In Multi-instruction Image Editing, we concatenate applicable instructions from MagicBrush's multi-turn dataset into a single instruction sequence. In the Context-Aware Instruction Image Editing task, we introduce non-applicable instructions generated with ChatGPT-4V(ision) [32] alongside images. More details on data creation are detailed in Section C.

**Metrics:** To evaluate our proposed method, we employ a diverse set of metrics, including L1/L2, CLIP-I, DINO, CLIP-T, CLIP-dir, and PickScore [22]. Detailed descriptions of these metrics are provided in Section B.

**Baselines:** We compare CAMILA with five different state-of-the-art image editing methods: IP2P [4], EMILIE [17], MGIE [10], SmartEdit (SE) [16], and FoI [11].

### 5.3 Main Results

**Quantitative Result.** As illustrated in Table 1, CAMILA demonstrates state-of-the-art results across both multi-instruction and context-aware instruction tasks, particularly excelling in metrics such as CLIP-I, CLIP-T, and DINO similarity, and overall distance metrics (L1 and L2). This indicates that our model aligns closely with human perception in maintaining fidelity to edited images.

The existing methods exhibit notable limitations. MGIE, which relies on a summarization approach to compress instructions, proves to be vulnerable to non-applicable instructions, leading to potential inaccuracies in execution. While SmartEdit shows improved understanding due to the integration of MLLMs, it suffers from a lack of robustness by feeding all instructions into the diffusion model simultaneously, which can lead to oversights in handling complex editing requests. Additionally, FoI struggles with imprecise attention maps, which reduces its performance below CAMILA though it is the most competitive baseline. In stark contrast, our approach effectively manages multi-instruction editing tasks, demonstrating superior capability in processing context-aware instructions.

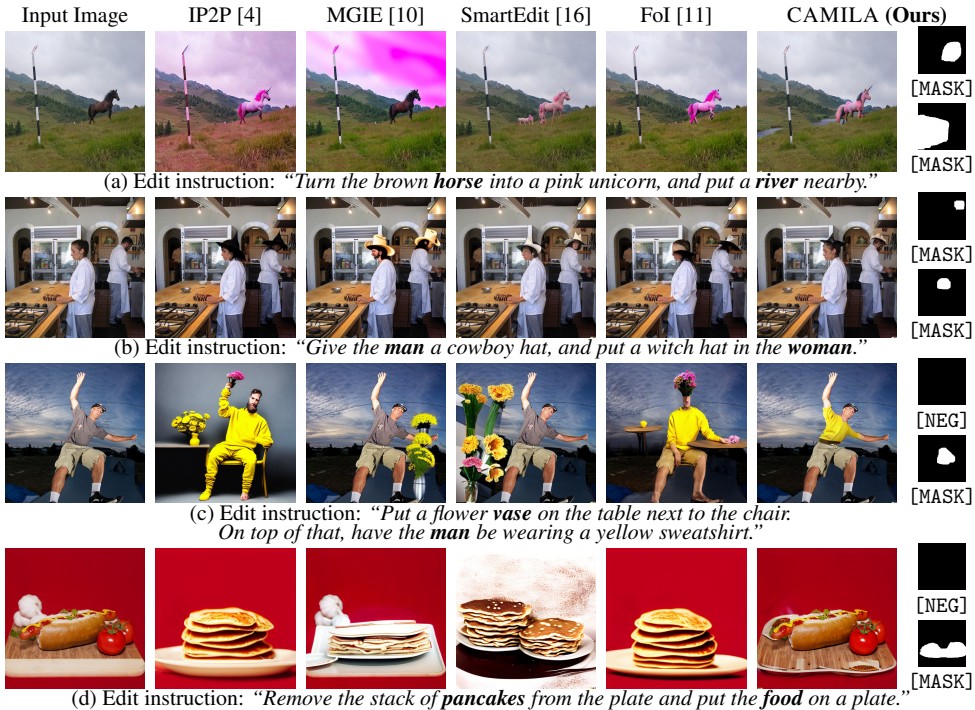

| Input Image | IP2P [4] | MGIE [10] | SmartEdit [16] | FoI [11] | CAMILA **(Ours)** |

(a) Edit instruction: *"Turn the brown **horse** into a pink unicorn, and put a **river** nearby."*

(b) Edit instruction: *"Give the **man** a cowboy hat, and put a witch hat in the **woman**."*

(c) Edit instruction: *"Put a flower **vase** on the table next to the chair. On top of that, have the **man** be wearing a yellow sweatshirt."*

(d) Edit instruction: *"Remove the stack of **pancakes** from the plate and put the **food** on a plate."*

Figure 4: **Qualitative comparisons:** FoI needs to extract keywords from each instruction using pretrained GPT model before running the model. Furthermore, due to inaccuracies in the attention map of diffusion model, FoI often fails to make precise modifications. In the case of context-aware instructions, CAMILA accurately identifies applicable instructions by generating [MASK] and [NEG] tokens from MLLM. We present the decoded mask results for each instruction of the [MASK] token.

This proficiency enables our model to execute edits with high precision, aligning closely with the intended modifications while minimizing the risk of over-editing. Overall, our results underscore the advantages of our method in navigating the complexities inherent to multi-instruction tasks.

In addition to distance-based metrics and similarity-based metrics, we further evaluate human perceptual alignment using the PickScore metric across two editing tasks: Multi-instruction and Context-Aware image editing. CAMILA outperforms the strongest baseline, FoI, by 18.1% and 24.0% in each setting, respectively. The advantage is more evident in the Context-Aware setting, demonstrating our model's ability to effectively filter non-applicable instructions. More detailed results are presented in Section E.1.

**Qualitative Result.** As shown in Figure 4, we present qualitative results and observe the following: All models, except for FoI, frequently execute only a single instruction when multiple instructions are provided. In (a), while FoI successfully performs the first instruction, it fails to generate an accurate attention map for the keyword 'river', resulting in the incomplete application of the second instruction. Similarly, in (b), the lack of a fine-grained attention map results in the hat being placed incorrectly. The remaining models predominantly execute only one instruction and demonstrate a tendency toward over-editing; for instance, IP2P and MGIE alter the background color in (a), and SmartEdit generates an additional unicorn.

In (c) and (d), most models exhibit erroneous edits in response to non-applicable instructions. In (c), despite the absence of a chair or table in the input image, the models add incorrect floral elements or a table. Similarly, in (d), although the input image does not contain a pancake, it erroneously appears due to the removal instruction, illustrating an inability to correctly handle the instruction. Especially compared to FoI, our model demonstrates greater precision in mask extraction for areas requiring modification, enabling more refined edits. Furthermore, CAMILA supports both localized object edits and global transformations. The [MASK] tokens dynamically adjust their spatial coverage based on each instruction, enabling edits that range from small regions to full-scene editings. Through the use of [MASK] and [NEG] tokens, our proposed model facilitates robust, context-aware image editing. Further qualitative results are provided in Section E.4.

Table 2: **Quantitative comparison on EMU dataset.** Achieving the highest CLIP-dir score in the Context-Aware task shows that our model effectively distinguishes non-executable instructions.

| Task | Single-inst | | Context-Aware | |
|---|---|---|---|---|
| Method | CLIP-T↑ | CLIP-dir↑ | CLIP-T↑ | CLIP-dir↑ |
| IP2P | 0.2616 | 0.075 | 0.2446 | 0.064 |
| MGIE | 0.2680 | 0.082 | 0.2543 | 0.066 |
| SE | 0.2680 | **0.094** | 0.2448 | 0.067 |
| FoI | 0.2673 | 0.068 | 0.2651 | 0.054 |
| **ours** | **0.2687** | 0.092 | **0.2679** | **0.092** |

Table 3: **Comparison of results before and after additional training with the surrogate module.** We apply the surrogate module to improve the CLIP-T score, which also enhances L1/L2 losses, as well as CLIP-I and DINO scores.

| Task | | Config | L1↓ | L2↓ | CLIP-I↑ | DINO↑ | CLIP-T↑ |
|---|---|---|---|---|---|---|---|
| Single Inst. | Single -Turn | before | 0.0602 | 0.0194 | 0.9367 | 0.9067 | 0.3020 |
| | | after | 0.0596 | 0.0191 | 0.9375 | 0.9069 | 0.3022 |
| | Multi -Turn | before | 0.0931 | 0.0339 | 0.8969 | 0.8357 | 0.3011 |
| | | after | 0.0782 | 0.0268 | 0.9127 | 0.8659 | 0.3019 |
| Multi Inst. | Multi | before | 0.0957 | 0.0372 | 0.8961 | 0.8329 | 0.2975 |
| | | after | 0.0945 | 0.0366 | 0.8980 | 0.8392 | 0.2984 |
| | Context -Aware | before | 0.0673 | 0.0228 | 0.9284 | 0.8910 | 0.3002 |
| | | after | 0.0661 | 0.0222 | 0.9296 | 0.8932 | 0.3006 |

# 6 Ablation Study

**Robustness of** CAMILA**.** In our framework, distinguishing applicable from non-applicable instructions is critical to prevent unintended edits. Each input may contain multiple instructions, which are classified by the MLLM into [MASK] and [NEG] tokens. On the Context-Aware Image Editing dataset, our model achieves a token classification accuracy of 90.21%, highlighting the robustness of CAMILA in filtering non-applicable instructions. Furthermore, we evaluate the alignment of generated masks with ground-truth edited regions using standard segmentation metrics. For applicable instructions, classified as [MASK] token, our model achieves an IoU of 0.3819 and a Dice score of 0.4986. As described in Equation (6), our model is trained with multiple loss objectives, not solely for segmentation accuracy. The generated masks are designed as high-level guidance, reflecting our focus on instruction fidelity and plausibility rather than strict spatial matching.

**Evaluation on Instruction-Following Accuracy.** We evaluate our method on the EMU dataset for both single-instruction and context-aware instruction tasks. Since the EMU dataset does not provide ground-truth target images, we utilize CLIP-T and CLIP-dir as evaluation metric. CLIP-dir measures how accurately the generated image aligns with the intended semantic direction of the instructions. As shown in Table 2, CAMILA achieves the highest CLIP-dir score in the context-aware instruction task, demonstrating its effectiveness in handling non-executable instructions.

Table 4: **Quantitative comparison on single instruction tasks.** CAMILA excels in single instruction tasks by generating precise masks that accurately target modification areas.

| | Single-turn Instruction | | | | | Multi-turn Instruction | | | | |
|---|---|---|---|---|---|---|---|---|---|---|
| Method | L1↓ | L2↓ | CLIP-I↑ | DINO↑ | CLIP-T↑ | L1↓ | L2↓ | CLIP-I↑ | DINO↑ | CLIP-T↑ |
| IP2P [4] | 0.1129 | 0.0373 | 0.8540 | 0.7423 | 0.2918 | 0.1538 | 0.0575 | 0.8103 | 0.6511 | 0.2866 |
| EMILIE [17] | 0.1129 | 0.0373 | 0.8540 | 0.7423 | 0.2918 | 0.1268 | 0.0509 | 0.8557 | 0.7591 | 0.2916 |
| MGIE [10] | 0.0931 | 0.0383 | 0.8853 | 0.8088 | 0.2935 | 0.1312 | 0.0574 | 0.8571 | 0.7507 | 0.3013 |
| SmartEdit [16] | 0.0895 | 0.0353 | 0.9030 | 0.8308 | **0.3024** | 0.1333 | 0.0575 | 0.8567 | 0.7421 | **0.3021** |
| FoI [11] | 0.0699 | 0.0206 | 0.9207 | 0.8779 | 0.2980 | 0.1084 | 0.0379 | 0.8681 | 0.7838 | 0.2935 |
| CAMILA (**ours**) | **0.0596** | **0.0191** | **0.9375** | **0.9069** | 0.3022 | **0.0782** | **0.0268** | **0.9127** | **0.8659** | 0.3019 |

**Single-Instruction Task Performance.** CAMILA, optimized for multi-instruction tasks, also performs strongly on single-instruction tasks, as shown in Table 4. CAMILA achieves strong performance across most evaluation metrics. In contrast, SmartEdit shows a tendency toward over-editing. This reflects CAMILA's balanced approach, minimizing over-editing while maintaining high fidelity. As shown in Figure 5, most models exhibit over-editing issues. Although FoI is designed to minimize over-editing, it encounters specific issues as follows: inaccurate attention maps in (a) prevent precise modifications to the 'angry birds' object, while in (b), additional frosting is incorrectly applied to cupcakes. These cases show that CAMILA achieves accurate edits without over-editing.

**Impact of Surrogate Module Training.** We compare the results on the multi-instruction tasks and single-instruction tasks before and after additional surrogate module training. As shown in Table 3, we demonstrate that mask generation through this surrogate module improves model performance. Interestingly, the improvement is not just for CLIP-T but also for the other metrics.

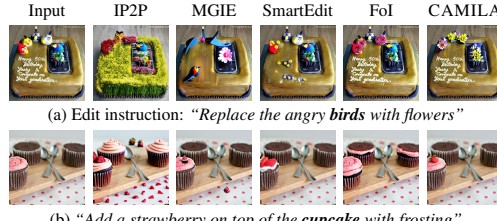

| Input | IP2P | MGIE | SmartEdit | FoI | CAMILA |

(a) Edit instruction: *"Replace the angry **birds** with flowers"*

(b) *"Add a strawberry on top of the **cupcake** with frosting"*

**Figure 5: Qualitative comparisons for single instruction task.** CAMILA demonstrates successful editing even in the single instruction task.

**Table 5: Large-scale baseline comparison on context-aware instruction task.** CAMILA outperforms larger diffusion-based models by generating contextually aligned and precise masks, highlighting that its context-aware design enhances editing quality without relying on increased model capacity.

| Method | L1↓ | L2↓ | CLIP-I↑ | DINO↑ | CLIP-T↑ |
|---|---|---|---|---|---|
| Step1X-Edit | 0.0830 | 0.0329 | 0.8096 | 0.8892 | 0.2984 |
| CAMILA | **0.0661** | **0.0222** | **0.9296** | **0.8932** | **0.3006** |

**Comparison with Large-Scale Baseline** We additionally compare CAMILA with recent large-scale model such as Step1X-Edit [28], which use larger diffusion backbone [34] than Stable Diffusion [38]. As shown in Table 5, CAMILA achieves better performance on the context-aware image editing dataset despite its smaller diffusion model size. This implies that the context-aware design of CAMILA effectively enhances editing fidelity without relying on large model capacity.

Additional ablation studies on the variation of the Token Decoder and the inference time comparison are provided in Section D.1 and Section D.2, respectively.

# 7 Conclusion

In this paper, we addressed the limitations of current text-guided image editing models, particularly their difficulty in handling fine-grained edits, multi-instruction edits, and distinguishing between executable and non-executable instructions. Leveraging MLLMs, we generated specialized tokens ([MASK] and [NEG]) and designed token broadcaster to ensure the validity of the contextual coherence between instructions and the image, so that only relevant edits can be applied to the designated regions while ignoring non-executable instructions. For comprehensive evaluation, we created new datasets that can evaluate Context-Aware Image Editing task, where our approach achieves superior results across both qualitative and quantitative evaluations compared to state-of-the-art solutions.

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
