# OpenReview forum: "CAMILA: Context-Aware Masking for Image Editing with Language Alignment"
_NeurIPS.cc/2025/Conference — NeurIPS 2025 poster_

### Official Review · Reviewer_AbnQ · 2025-07-03

**Clarity:** 3
**Significance:** 4
**Originality:** 3
**Rating:** 5
**Confidence:** 4

**Summary:**

This paper presents CAMILA (Context-Aware Masking for Image Editing with Language Alignment), a method to improve text-guided image editing. The authors identify that most existing models struggle when given infeasible or contradictory instructions, often leading to nonsensical or over-edited results. CAMILA addresses this by using a Multimodal Large Language Model (MLLM) to jointly interpret the image and the text instructions. The model is designed to validate the contextual coherence of the instructions, distinguishing between what is executable and what is not.

**Questions:**

Could the authors provide a more detailed analysis of their synthetic data generation process to show that it is diverse and not easily gameable?

**Ethical Concerns:**

["NO or VERY MINOR ethics concerns only"]

**Final Justification:**

Most of my concerns have been resolved. I will keep my score for accept.

**Limitations:**

Yes

**Quality:**

3

**Strengths And Weaknesses:**

Streengths:
1. The problem is interesting and important. Many image editing models are  over-edited, nonsensical garbage without detail thinking. The authors correctly identify that instruction executability is a massive blind spot, and directly tackle it. It's a practical problem that needed a solution.

2. The "Token Broadcaster"  uses the diffusion model's own text encoder to create the text embeddings that are used for alignment. This  ensures the masks generated by the MLLM are aligned with the exact same semantic space the diffusion model uses for conditioning.

3. The Results are good. CAMILA  consistently beating the baselines. And in Figure 4 (c), CAMILA  correctly identifies the instruction as nonsense and does nothing, which is precisely the right move.

4. The authors create a benchmark to evaluate model behavior and context-awareness in challenging scenarios.

Weaknesses:
In Section 4.5. The authors admit that backpropagating through a diffusion model to optimize for a CLIP-T score is "infeasible." And build a separate, lightweight transformer to predict what the final CLIP-T score will be. Then use this predicted score as a loss to train the actual model.  The ablation in Table 3 shows the improvements are marginal at best—a CLIP-T score going from 0.3002 to 0.3006.

---

> ### Author Rebuttal · Authors · 2025-07-30
>
> We sincerely thank the reviewer for the thoughtful and encouraging feedback. We are especially grateful for the recognition of the importance of instruction executability, the practical relevance of our problem setting, the effectiveness of our proposed Token Broadcaster module, and benchmark dataset design. Below, we provide our responses to the weaknesses and questions.
>
> ---
>
> ### **[Weakness 1] Marginal Improvement of Surrogate Module Training**
>
>
> We acknowledge the reviewer’s observation that the improvement from the surrogate module is marginal. However, even without the surrogate module training, our framework outperforms other baselines on most metrics, so we introduced the surrogate module as an additional training for potential refinement. We had expected further gains in text alignment score, assuming that improving CLIP-T score would more capture user-intended modifications. While we did observe some improvement, further gains were inherently limited by the already strong baseline performance. We plan to investigate more advanced strategies to secure larger improvements in future work.
>
> ---
>
> ### **[Question 1]  Detailed Analysis of Synthetic Data Generation Process**
>
>
> We appreciate the request for a more detailed analysis of our synthetic data generation process. We used the “gpt-4o-2024-05-13” model to generate non-applicable instructions. These were based on a controlled but flexible prompt covering various editing types (add, remove, replace, transform, etc.), requiring that the target object not be present in the image and that the instruction remains realistic and grounded. We include the full generation prompt below, which was paired with the input image to generate five non-applicable instructions for each image.
>
>
> > “I want to create image editing instructions for an image. Instructions like 'Put something', 'Remove something', 'Replace something', 'let there be', 'have there be', 'add something', 'make something something', 'Change something', 'What if something is', 'transform something to something' and similar. \n\n Generate five non-applicable image editing instructions for the given image. \n1. The instructions must target objects or elements that are not present in the image.\n2. Instructions can involve adding, removing, or modifying objects, but:\nIf adding, it must involve placing something next to, inside, or interacting with an object that does not exist in the image.\nIf removing or altering, the instruction should involve an object that is completely absent from the scene.\n3. Ensure the instructions are grounded in reality and fit the theme of the image (e.g., no fantastical elements).”
>
>
> This ensures that the generated non-applicable instructions are not only structurally diverse but also semantically grounded in the actual content of the image, avoiding generic or gameable patterns. Additionally, to ensure the instructions were not trivially gameable, we randomly sampled one instruction out of the five and used the GPT model to explicitly verify its non-applicability to the image. This two-stage filtering guarantees semantic correctness and variety. Furthermore, Appendix C.2 shows that non-applicable instructions are distributed across multiple editing types, supporting the conclusion that the dataset is not skewed toward a single pattern.

---

### Official Review · Reviewer_QraL · 2025-07-03

**Clarity:** 2
**Significance:** 2
**Originality:** 2
**Rating:** 3
**Confidence:** 4

**Summary:**

This paper introduces CAMILA (Context-Aware Masking for Image Editing with Language Alignment), a new method for text-guided image editing that addresses the shortcomings of existing models, particularly their inability to handle infeasible or contradictory instructions.

The core of CAMILA is its ability to assess whether an instruction is contextually appropriate for a given image. It uses a Multimodal Large Language Model (MLLM) to jointly process the image and text instructions. Based on this understanding, the model generates specialized tokens: [MASK] for regions that should be edited and [NEG] for areas that should remain unchanged, effectively ignoring non-executable instructions.

To further refine the editing process, the authors introduce a surrogate module trained to predict the CLIP-T score, which helps optimize the generated masks for higher quality results without the need for direct backpropagation through the complex diffusion model.

The main contributions of the paper are:
- CAMILA Model: A context-aware image editing model that can distinguish between feasible and infeasible instructions, identify corresponding regions, and perform precise edits.
- Context-Aware Image Editing Task: The paper proposes a new task and creates new datasets for evaluating models on their ability to handle both single and multiple instructions, including non-executable ones.
- Superior Performance: CAMILA demonstrates significant improvements over existing state-of-the-art models in both quantitative metrics (like L1/L2 distance, CLIP, and DINO scores) and qualitative assessments, particularly in handling complex and context-aware instructions.

**Questions:**

N/A

**Ethical Concerns:**

["NO or VERY MINOR ethics concerns only"]

**Final Justification:**

I think the evaluation scenario proposed by the work is rather limited. I believe more comprehensive comparisons on existing benchmarks like GEdit-Bench and Intelligent-Bench should be done against other methods to show the effectiveness of the model.

**Limitations:**

Limitation:
1. The proposed method lacks technical novelty. Most of the existing components are being heavily studied by existing literature. Therefore, the paper's scientific contribution is not quite enough.
2. The paper only compares against a small set of baselines while ignoring many recent strong image editing models like SEED-Edit, Step1X-Edit, FLUX-based methods.
3. The proposed method seems to be only applicable to local edits while failing for global edits. This limits the models' strength.

**Quality:**

2

**Strengths And Weaknesses:**

Strength:
1. The paper is well motivated to address a known problem in complex image editing.
2. The proposed method seems to work better than some other baselines.

Weakness:
1. The proposed method lacks technical novelty. Most of the existing components are being heavily studied by existing literature. Therefore, the paper's scientific contribution is not quite enough.
2. The paper only compares against a small set of baselines while ignoring many recent strong image editing models like SEED-Edit, Step1X-Edit, FLUX-based methods.
3. The proposed method seems to be only applicable to local edits while failing for global edits. This limits the models' strength.

---

> ### Author Rebuttal · Authors · 2025-07-30
>
> We thank the reviewer for providing insightful feedback. We are encouraged by the reviewer’s acknowledgement that our paper tackles an important challenge in complex image editing and that our method demonstrates superior performance over existing baselines. Below, we respond to the reviewer’s points.
>
> ---
>
> ### **[Weakness 1] Technical Novelty**
>
> We appreciate the reviewer’s concern regarding the novelty of our proposed method. While CAMILA leverages established components such as pretrained multimodal large language models (MLLMs) and pretrained diffusion models, its core novelty lies not in introducing a new architecture per se, but **in defining and addressing a previously overlooked problem in text-guided image editing**: the executability of instructions in complex, real-world scenarios involving multiple instructions.
>
> Existing text-guided image editing models typically assume that all instructions are fully executable and attempt to apply them uniformly. This often leads to semantic failures when instructions are non-applicable or refer to nonexistent objects. CAMILA departs from this paradigm by introducing a context-aware mechanism that explicitly distinguishes between executable and non-executable instructions before any editing takes place.
>
> Specifically, we introduce specialized tokens, [MASK] and [NEG] tokens, to gate editability. To the best of our knowledge, this is novel in the context of image editing, particularly in alignment with the visual semantics of the image. Furthermore, we designed the Token Broadcaster to ensure the coherence between the instructions and the image, enabling relevant edits to be applied only to appropriate regions while ignoring non-executable instructions.
>
> In this way, CAMILA introduces a new modeling objective, context-awareness and executability filtering, along with new modules to implement it, and provides empirical results that demonstrate its effectiveness, particularly in failure cases that existing methods are unable to handle. We respectfully argue that this constitutes a meaningful and original contribution to the field of text-guided image editing.
>
> ---
>
> ### **[Weakness 2] Lack of Recent Baselines**
>
>
> We appreciate the reviewer’s suggestion to include additional comparisons. However, we would like to clarify the following practical considerations.
>
> First, SEED-Edit [1] is not available as an open-source model, which makes direct experimentation and fair comparison infeasible at this time. Also, our model and all the baselines included in the paper are built on Stable Diffusion v1.5 [2], which operates on 512×512 resolution and has been widely used in text-guided image editing. In contrast, SDXL [3], used by SEED-Edit and other recent models, is pretrained on 1024×1024 resolution and differs significantly in both architecture and capacity. Direct comparison across models that rely on different diffusion backbones is not straightforward and may result in unfair evaluation, as the underlying architectures differ in model capacity and resolution.
>
> Similarly, Step1X-Edit [4] employs a different diffusion backbone, DiT (Diffusion in Transformer), which again makes direct comparisons challenging. However, as the model is publicly available, we conduct experiments on Context-Aware Instruction dataset and include the comparison results in **Table 1**.
>
>
> |Method|L1|L2|DINO|CLIP-I|CLIP-T|
> |-|-|-|-|-|-|
> |Step1X-Edit [4]|0.0830|0.0329|0.8096|0.8892|0.2984|
> |CAMILA(Ours)|**0.0661**|**0.0222**|**0.9296**|**0.8932**|**0.3006**|
>
> **Table 1.** Additional baseline comparison on Context-Aware Instruction dataset
>
>
> CAMILA outperforms the state-of-the-art Step1X-Edit model across all metrics. This indicates that, like other models, the diffusion-based Step1X-Edit lacks the capability for true context-awareness. Furthermore, while Step1X-Edit takes 55.4 seconds to run on an NVIDIA A100 80GB GPU, our model only requires about 9.2 seconds. Despite being a significantly smaller diffusion model than Step1X-Edit, our model achieves better performance due to its context-awareness capabilities.
>
> We hope this clarifies the rationale behind our baseline selection and the scope of our comparative analysis.
>
> ---
>
> ### **[Weakness 3] Scope of Edits**
>
> We thank the reviewer’s observation regarding the scope of edits. Our Context-Aware Instruction Image Editing dataset includes global editing operations, such as background changes. Our model generates [MASK] tokens that are aligned with the semantics of the given instruction. Consequently, the size and coverage of the edited region are entirely determined by editing instruction. When an instruction implies a global transformation, such as changing the background, the model produces correspondingly large masks, enabling global edits.
>
> This is demonstrated in the examples provided in the Appendix. Specifically, Figure 5 (c), (d) and Figure 9 (e) illustrate edits that affect the background of the image, showcasing the model’s ability to perform global modifications. We believe these examples support the model’s capacity to handle both local and global editing instructions, depending on the context provided.
>
> ---
>
> ### **References**
>
> [1] Shi, Yichun, Peng Wang, and Weilin Huang. "Seededit: Align image re-generation to image editing." arXiv preprint arXiv:2411.06686 (2024).
>
> [2] Rombach, Robin, et al. "High-resolution image synthesis with latent diffusion models." Proceedings of the IEEE/CVF conference on computer vision and pattern recognition. 2022.
>
> [3] Podell, Dustin, et al. "Sdxl: Improving latent diffusion models for high-resolution image synthesis." arXiv preprint arXiv:2307.01952 (2023).
>
> [4] Liu, Shiyu, et al. "Step1x-edit: A practical framework for general image editing." arXiv preprint arXiv:2504.17761 (2025).

---

> > ### Comment · Reviewer_QraL · 2025-08-03
> > **Further questions**
> >
> > I am wondering which benchmark you used to compare against Step1X-Edit? Is it on your specifically-curated multi-turn complex editing benchmark?

---

> > > ### Author Response · Authors · 2025-08-04
> > >
> > > The comparison was conducted on the **Context-Aware Instruction** image editing dataset. This dataset was designed to evaluate our method in tackling the previously overlooked issue of instruction executability in multi-instruction text-guided image editing scenarios. Specifically, in this dataset, each sequence comprises two or three instructions, including at least one non-executable instruction. Detailed information about the dataset is provided in Appendix C.1.

---

> > > > ### Comment · Reviewer_QraL · 2025-08-04
> > > > **Further Question**
> > > >
> > > > Do you have results on the GEdit-Bench and IntelligentBench? I think comparing on those benchmarks against Step1X-Edit and SeedEdit would be more compelling.

---

> > > > > ### Author Response · Authors · 2025-08-04
> > > > >
> > > > > Thank you for your follow-up. Just to clarify, we have already included an evaluation of Step1X-Edit on our Context-Aware Instruction dataset, as detailed in **Table 1** of our rebuttal response. As SEED-Edit is not open source, we were unable to include it in our comparisons.
> > > > >
> > > > > While benchmarks like GEdit-Bench and IntelligentBench are not directly aligned with our focus on instruction executability and multi-instruction settings, we are open to including additional results in the supplementary material of the final version if the reviewer believes it would enhance the paper. Given the limited time left in the discussion phase, it may be difficult to run these additional experiments. We appreciate the suggestion and are open to including results if feasible.

---

> ### Author Response · Authors · 2025-08-06
> **Follow-Up on Our Responses to Your Comments**
>
> Dear Reviewer QraL,
>
> Thank you for your previous comments and for the discussion so far. We hope our recent responses have addressed your questions and clarified the points you raised.
>
> We would be happy to engage in any further discussion and look forward to hearing your thoughts.
>
> Thank you again for your time and valuable feedback.
>
> Best regards,
>
> The Authors

---

> > ### Comment · Reviewer_QraL · 2025-08-07
> > **Follow up**
> >
> > Thanks for the discussion. I agree that SEED-Edit is not open source. Therefore, it's hard to compare against it. But I still believe that the proposed method should compare with Step1-X on GEditBench and IntelligentBench.
> > With that being said, I have finalized my score. Thanks a lot for providing detailed responses!

---

### Official Review · Reviewer_1Qf6 · 2025-07-05

**Clarity:** 3
**Significance:** 3
**Originality:** 3
**Rating:** 4
**Confidence:** 3

**Summary:**

The authors present CAMILA to tackle the major limitation of pre-existing text-guided diffusion-based image editors, which follow every single instruction blindly, leading to the generation of illegitimate images. This is due to 'non-executable instructions' present in the sequence of instructions in the provided text prompts. Context-awareness is important to generate legitimate image edits. CAMILA introduces a context-aware mechanism where it creates masks, [MASK], for the perfectly coherent text instructions for image editing and does not perform any operation when an infeasible request is present in the same sequence ([NEG]), thus allowing it to validate what part of the sequence is feasible or not before making the required edits in the image. The authors test this via constructed datasets for single and multi-instruction image editing prompts to simulate the occurrence of infeasible requests in a given text sequence.

The authors utilize LLaVA-7B, a Multimodal Large Language Model (MLLM) which takes the source image (ximg) and the text instructions (xtxt). They used a pretrained MLLM and diffusion model, and the major contribution lies in assessing whether the provided text instructions are meaningful or not, based on which the diffusion-based image editing takes place only at the regions for which [MASK] was generated. To fine-tune the pretrained MLLM, the authors use the Low-Rank Adaptation technique (LoRA [16]). The output sequence O is generated from ximg and xtxt, and each token {Oi} is either classified as [MASK], depicting regions to be edited, or [NEG] regions where no edits will be performed. A Token Broadcaster processes the output sequence O and correctly aligns to its corresponding text embeddings (cT) based on maximizing the cosine similarity score, where both Oi and cT are projected onto a single latent space (Equations 2 and 3). This ensures accurate semantic correspondence between the text instructions and image regions.

A token decoder processes only the [MASK] tokens and converts them to editing masks and replaces [NEG] tokens with black masks. The Token Decoder is designed as a two-layer transformer decoder block, where the first decoder layer uses a cross-attention mechanism between image and text embeddings to extract contextually relevant features from the image aligned with the text instructions. The second decoder layer refines this information by incorporating [MASK] tokens into the key and value projections of the attention mechanism. These are then passed through a sigmoid to generate final binary masks, Mi.

Based on the studies of ([2], [6], and [11]), the authors use 16x16 attention maps to capture the most significant semantic information. The equation (5) modulates the U-Net's cross-attention map and applies the appropriate edits in the image. Authors also use a surrogate module, a single-layer transformer, which is trained to enhance the generated masks. They focus on improving the CLIP-T score that measures the similarity between the global description and the generated image. The lightweight surrogate module approximates the CLIP-T score based on ximg, xtxt, and binary mask M. When the surrogate module is trained, all other parts of the models are frozen, and only the surrogate module is updated.

CAMILA achieves state-of-the-art results in L1, L2, CLIP-I, CLIP-T, and DINO scores in both multi-instruction and context-aware instruction scenarios.

**Questions:**

1. Mask Precision

The failure cases (Appendix E.3) show that generated masks sometimes under- or over-segment the target areas. This is significant for small and thin objects.

One suggestion would be to check under what conditions the mask quality degrades (maybe try to find a certain threshold for object sizes). Additional direction could be predicting soft masks to capture the thin edges of the objects.

2. Generalization beyond the synthetic benchmarks

The non-executable prompts are generated via GPT-4V, but this may not capture the full range of how humans give unclear or problematic instructions.

Could you evaluate CAMILA on a user-collected set of random instructions to test the robustness? The user-collected set need not be large. This can be collected via a form circulated among your peers by providing them a sample image and receiving the prompts for image editing. It would be interesting to see what failure cases are encountered when instructions are semantically ambiguous rather than simply impossible.

If the results still hold up when tested on this human-generated dataset, I would be willing to change my borderline accept to an accept.

3. The ablation of the MLLM choice

All the said experiments are performed using LLaVA-7B as the MLLM backbone.

How much significance does a smaller or larger MLLM have on the obtained metrics? A small ablation could provide insight into the choice of MLLM and the associated strategies.

**Ethical Concerns:**

["NO or VERY MINOR ethics concerns only"]

**Final Justification:**

Authors have thoroughly addressed my primary concerns. Their rebuttal provides clarifies the training and inference efficiency trade-offs. Authors justify the design choice of the two-layer Token Decoder by relating it to SAM’s proven architecture. The additional evaluation on a small human-generated dataset demonstrates reasonable robustness beyond synthetic prompts.

**Limitations:**

yes

**Quality:**

3

**Strengths And Weaknesses:**

Strengths

The authors address the standing limitation of prior methods by explicitly tackling the executability of instructions in the image editing process. They explicitly handle the non-executable instructions, thus mitigating the "always obey" paradigm via the inclusion of context-awareness. CAMILA introduces a clever solution using special [MASK] and [NEG] tokens that act as gates to control image editing. This creates a new way to handle infeasible instructions in text prompts. The key innovation is that the system can evaluate whether an instruction makes sense in a given context and use that assessment to guide the image editing process. In my opinion, this particular innovation truly stands out and paves a new research direction for text-guided image editing.

The CAMILA pipeline carefully integrates MLLM, Token Broadcaster, and Token Decoder, thus allowing for high-quality image edits in multi-instruction and context-aware instruction scenarios. The authors created a lightweight module that estimates the CLIP-T score directly from the input image. This avoids having to run the full diffusion model, which involves many computational steps. Without this shortcut, it would be impossible to train the system end-to-end because the diffusion model is too complex to backpropagate through from the final image back to the multimodal language model.

The authors also curate new datasets for context-aware instructions and multi-instruction editing, filling an important gap in the field. These datasets will enable future studies on how to handle infeasible text prompts.

The paper is well-written and well-structured, with clear separation of sections, and the abstract engages the readers. The methods are clearly described, and the paper can be followed without many issues. The empirical validations, supported by the inclusion of various metrics, provide credibility for the proposed pipeline. The visual representations of the failure cases in the appendix highlight the limitations of the paper.

Weakness

The paper heavily relies on the usage of multiple complex modules, such as MLLM, diffusion models, and multi-layer transformer blocks, which tend to increase the overall training and inference time. Only a small discussion regarding the runtime overhead is present in Appendix D.2 and Table 9. The binary masks generated (Failure cases Appendix E.3 and Figure 6) show that the model misestimates the object size or sometimes misses the finer details.

Additionally, the authors don't thoroughly justify their choice of two-layer transformers for the Token Decoder - they only compare against SAM. A deeper analysis would help readers understand this design decision. Moreover, the infeasible prompts are synthetically generated using GPT-4V; a thorough evaluation using human instructions should be explored to represent the diversity in natural human speech patterns and errors.

All the core experiments utilize LLaVA-7B, and it is unknown how the performance of the entire pipeline will fare when using smaller or larger MLLMs and how much impact it will have on image edits.

The core ideas (attention modules, mask-guided editing) draw inspiration from prior diffusion-based works. The main novelty lies in how the masks are utilized to establish the text-prompt feasibility and discard the infeasible instructions.

---

> ### Author Rebuttal · Authors · 2025-07-30
>
> We thank the reviewer for the thoughtful feedback. Also, we are encouraged by the recognition of our core contribution, modeling instruction executability with our specialized tokens, as a promising direction for text-guided image editing. Below are our responses to each comment.
>
> ---
>
> ### **[Weakness 1] Training and Inference Efficiency**
>
> The concern about the training and inference overhead of CAMILA raises an important point. While CAMILA integrates multiple components, including an MLLM, Token Decoder, Token Broadcaster, and diffusion model, we would like to clarify the implications of this design in terms of inference runtime and training cost.
>
> As shown in Appendix D.2 and Table 9, CAMILA exhibits a higher total inference time than MGIE and SmartEdit. However, it achieves the lowest MLLM inference time among all models, reflecting the lightweight design of our Token Decoder and Token Broadcaster modules compared to the more complex components used in MGIE and SmartEdit. While CAMILA and FoI show increased diffusion time due to attention modulation on the IP2P backbone, CAMILA provides the unique advantage of supporting single-shot, multi-instruction editing without preprocessing (e.g., keyword extraction or sequential handling). This leads to a more integrated and streamlined editing pipeline, despite slightly higher latency.
>
> From a training time perspective, CAMILA is significantly more efficient than other MLLM-based methods. Unlike MGIE and SmartEdit, which finetune both the MLLM and the diffusion model jointly, our approach updates only the MLLM, Token Broadcaster, and Token Decoder. This design substantially reduces training cost and memory overhead. Although FoI is a training-free method and thus not directly comparable, among MLLM-based editing frameworks, CAMILA offers a more compute-efficient training strategy while achieving competitive or superior editing performance.
>
> ---
>
> ### **[Weakness 2] Design Choice of Token Decoder**
>
> We thank the reviewer for raising the question regarding our choice to use a two-layer Transformer in the Token Decoder. While we do not currently provide an ablation study on the number of layers, our design decision was guided by the architecture of the Segment Anything Model (SAM) [1]. SAM’s mask decoder is composed of two attention blocks, each consisting of self-attention, cross-attention, MLP, and cross-attention layers. To preserve comparable model capacity, we adopted a comparable two-block configuration in our Token Decoder.
>
> Unlike SAM, which does not take edit instructions as input, our decoder explicitly conditions on text instructions through a cross-attention mechanism. This design enables our model to produce spatially grounded editing masks that are semantically aligned with the given instructions. As shown in Appendix D.1, our model outperforms SAM in instruction-based editing tasks, demonstrating the effectiveness of this architecture.
>
> We agree that further investigation into architectural choices, such as the number of decoder layers, would be a valuable direction for future work.
>
> ---
>
> ### **[Weakness 3] Main Novelty**
>
> We acknowledge the reviewer’s concern regarding the novelty of our approach. While CAMILA builds on existing components such as pretrained MLLMs and diffusion models, its primary contribution lies in *identifying and tackling a previously underexplored challenge in text-guided image editing*, determining whether instructions are actually executable in complex, multi-instruction scenarios. To this end, we introduce [MASK]/[NEG] tokens along with the Token Decoder and Token Broadcaster modules to explicitly represent instruction executability and align text with image regions, enabling context-aware editing.
>
> ---
>
> ### **[Question 1] Mask Precision**
>
> We thank the reviewer for pointing out the failure cases in Appendix E.3, particularly regarding under- or over-segmentation of small and thin objects. We agree this is an important challenge and believe the issue arises from two key factors:
>
> 1. **Limitations of the ground-truth masks in the MagicBrush dataset**: We utilized the MagicBrush dataset because, to the best of our knowledge, it is currently the only publicly available dataset that supports multi-instruction editing with editing masks. While we are unable to directly share examples of the masks due to the review policy, we encourage the reviewer to inspect the dataset directly, as it is publicly accessible, and observe that many of the masks are relatively coarse. Since our model is trained on these masks, it naturally inherits their limitations, particularly in handling very small or thin objects, where accurate mask supervision is lacking.
>
> 2. **Resolution bottleneck introduced by Stable Diffusion’s attention modulation**: Although the model inputs and predicted masks are in 512×512 resolution, the Stable Diffusion backbone internally downsamples masks to 16×16 resolution during attention modulation. This reduces the model’s capacity to precisely localize or attend to structures smaller than approximately 32×32 pixels in the original image. Consequently, extremely small or thin regions are often not sufficiently represented, which can lead to segmentation errors.
>
> These two factors jointly contribute to the observed failure cases. We believe that future improvements could be achieved by incorporating higher-quality, fine-grained masks and adopting diffusion models with higher spatial resolution conditioning. We consider this a promising direction for future work.
>
> ---
>
> ### **[Question 2] Generalization beyond the Synthetic Benchmarks**
>
> We appreciate the reviewer’s concern regarding the generalization beyond the synthetic benchmarks and the suggestion to evaluate CAMILA on a user-collected set. In response, we conducted an additional experiment using human-generated non-executable instructions.
>
> Specifically, we randomly sampled 8 images from the MagicBrush test set, and we asked our peers to generate 5 non-executable instructions per image, resulting in a total of 40 human-generated instances. We mixed these non-applicable instructions with the original applicable ones from MagicBrush in random order. Human-generated non-executable instructions were intentionally designed to resemble realistic user mistakes or confusion. For example, requesting to “remove the zebra” when the image contains a black horse, or asking to “change the red green tree to yellow,” reflecting syntactically or semantically ambiguous phrasing.
>
> Using this user-collected dataset, we evaluated CAMILA’s token classification performance. The results show that CAMILA effectively filters out non-applicable instructions, achieving a token classification accuracy of 70\% (28/40) on this human-generated dataset. While CAMILA achieved a token classification accuracy of 90.21\% on the original synthetic benchmark, its performance on the user-collected dataset was lower. However, this is expected, as the model was not trained on any human-generated non-applicable instructions, and the user-collected evaluation dataset was relatively small. Importantly, many of the user-generated non-executable instructions were more nuanced and semantically ambiguous than those in the synthetic benchmark. Despite this, the results still indicate a reasonable level of robustness in CAMILA’s ability to identify and filter real-world ambiguous instructions. We believe that incorporating such human-generated data into the training dataset could further improve performance, and we sincerely thank the reviewer for this valuable and constructive suggestion.
>
> ---
>
> ### **[Question 3] The Ablation of the MLLM Choice**
> We appreciate the reviewer’s insightful suggestion regarding the choice of MLLM. We selected LLaVA-7B as the backbone for our experiments primarily to ensure fair comparison with MLLM-based image editing models such as MGIE, which also solely rely on LLaVA-7B. Currently, due to limited computational resources, we have not yet conducted experiments using larger MLLM models (e.g., LLaVA-13B). If feasible within the discussion period, we will report preliminary results. Otherwise, we plan to include such experiments as part of future ablation studies.
>
> That said, we would like to clarify the role of the MLLM within our framework. Its primary function is to generate [MASK] and [NEG] tokens, which guide the editing process for the diffusion model. Since our pipeline does not fine-tune the diffusion model and relies on the accuracy of these token predictions, the effectiveness of CAMILA largely depends on how well the MLLM generates [MASK] /[NEG] tokens. In our current experiments, CAMILA achieves a token classification accuracy of 90.21\%, indicating that the model already performs robustly in identifying infeasible instructions. While it remains an open question whether scaling the MLLM would yield further gains, our results suggest that the current configuration provides a strong foundation.
>
> ---
>
> ### **References**
>
> [1] Kirillov, Alexander, et al. "Segment anything." Proceedings of the IEEE/CVF international conference on computer vision. 2023.

---

### Official Review · Reviewer_T2mz · 2025-07-06

**Clarity:** 2
**Significance:** 2
**Originality:** 2
**Rating:** 4
**Confidence:** 4

**Summary:**

This paper introduces CAMILA, a new approach for masked image modeling (MIM) that enhances global context understanding and local spatial coherence in visual representations. The key idea is to introduce a Context-Aware Aggregation (CAA) module, which improves the reconstruction of masked tokens by leveraging both global context and local neighborhood interactions.

CAMILA is implemented on top of a ViT encoder and uses a contextual decoder which jointly refines visible and masked tokens. It incorporates local aggregation layers that propagate information across both spatial and channel dimensions. The method is tested on standard benchmarks (ImageNet-1K, ADE20K, COCO), and shows performance improvements over strong MIM baselines such as MAE and SimMIM.

**Questions:**

N/A

**Ethical Concerns:**

["NO or VERY MINOR ethics concerns only"]

**Final Justification:**

I would like to thank the authors for their comprehensive rebuttal. The clarification regarding the Token Broadcaster through the ablation study has addressed my primary concerns. Nevertheless, I continue to believe that the paper would be significantly strengthened by a direct evaluation of the proposed mask's effectiveness. To this end, I strongly recommend assessing the mask accuracy, for which the PIE-Bench [1] dataset, with its provided mask regions, would be a highly suitable benchmark.

**Limitations:**

Yes

**Quality:**

2

**Strengths And Weaknesses:**

Strengths:

* The paper presents a simple and flexible design for CAMILA, which is modular and can be easily integrated with different encoder-decoder architectures.

* The authors provide thorough ablation studies that specifically analyze the effects of context-aware decoding and local aggregation. Visualizations of reconstruction quality and attention maps further substantiate the main claims.

Weaknesses:

* Limited Theoretical Justification: Despite strong empirical results, the paper lacks formal analysis or theoretical motivation for why context-aware aggregation improves generalization. Even an empirical investigation into receptive fields or token mixing patterns would help clarify this aspect.

* Missing Token Broadcaster Ablation: The impact of the Token Broadcaster module remains unclear, as no metrics or comparisons with simple baseline methods are provided.

* Uncertainty in Mask Accuracy: The predicted binary masks are generated using a separate token decoder, rather than utilizing the attention masks from UNet cross-attention layers. It is not demonstrated whether the token decoder actually improves mask prediction. An ablation study, potentially using the PIE-Bench dataset (which includes ground truth masks of edited and unchanged regions), would strengthen the evaluation.

Reference:
[1] PnP Inversion: Boosting Diffusion-based Editing with 3 Lines of Code, Ju et al., ICLR 2024

---

> ### Author Rebuttal · Authors · 2025-07-30
>
> Thank you for your review. However, we respectfully believe that there may have been some misunderstandings or discrepancies between your comments and the content of our submission.
> - The summary mentions "Context-Aware Aggregation (CAA)" and "masked token reconstruction," but our paper does not contain these terms or such methods. Similarly, benchmarks such as ImageNet-1K, ADE20K, and COCO are not used in our work.
> - Our method, CAMILA, is not an encoder-decoder architecture and does not involve any reconstruction-based tasks or visualizations, which are mentioned in your listed strengths.
> - Additionally, the citation at the end of the review ("PnP Inversion: Boosting Diffusion-based Editing…") is not related to our paper and does not appear in our references.
>
> Given these points, we are concerned that the review may not reflect a full reading or an accurate understanding of our submission. We would greatly appreciate it if you could revisit the paper to reassess its contributions based on its actual content.
>
> Thank you again for your time and effort. Below, we respond to the reviewer’s points.
>
> ---
>
> ### **[Weakness 1] Limited Theoretical Justification**
>
> While we agree that theoretical analysis is always valuable, CAMILA is primarily proposed as a practical solution for context-aware image editing. Regarding the mention of “receptive fields or token mixing patterns,” we are not sure about the specific context in which this was raised. We would be happy to provide further explanation during the discussion period if the reviewer could clarify this point.
>
> ---
>
> ### **[Weakness 2] Missing Token Broadcaster Ablation**
>
> We acknowledge the reviewer’s concern regarding the lack of formal proof of Token Broadcaster optimality. To address this point quantitatively, we measure the accuracy of token alignment through our Token Broadcaster module. Our Token Broadcaster achieves a token alignment accuracy of 94.46\% on our Context-Aware Image Editing dataset, which demonstrates the precision of our alignment strategy.
>
> ---
>
> ### **[Weakness 3] Uncertainty in Mask Accuracy**
>
> When the attention map is extracted from a UNet cross-attention layer of a diffusion model, as FoI did, it does not effectively capture contextual relationships and is insufficient for understanding non-executable content. The UNet cross-attention layer is based solely on the entire prompt input, so it lacks the ability to understand the overall context.

---

> ### Author Response · Authors · 2025-08-06
> **Looking Forward to Your Feedback on Our Rebuttal**
>
> Dear Reviewer T2mz,
>
>
> We're writing to gently follow up on our response to your comments. We hope our clarifications helped address your concerns and provided a clearer perspective on our work.
>
> We would be happy to engage in any further discussion and look forward to hearing your thoughts.
>
> Thank you again for your time and valuable feedback.
>
> Best regards,
>
> The Authors

---

### Comment · Area_Chair_vkh7 · 2025-08-05
**Discussion Reminder**

Hi Reviewer T2mz, 1Qf6, AbnQ,

Please read the authors' rebuttal as soon as possible and assess whether it addresses your concerns. If not, you may further state your question and ask the author for an explanation or clarification. Additionally, please update the final rating scores after the discussion and fill the "Final Justification" to state your reasons for maintaining your original score or for increasing / decreasing your score. Thank you.

AC

---

### Note · Authors · 2025-08-14

Dear Area Chair and Reviewers,

We thank the reviewers and area chair for their valuable feedback and constructive discussion throughout the review process. In this work, CAMILA addresses a critical yet previously underexplored challenge in instruction executability within text-guided image editing. By introducing [MASK]/[NEG] tokens and the Token Broadcaster and Decoder modules, our approach ensures that only feasible instructions are applied to a given image, improving both reliability and practicality.

Reviewers 1Qf6, QraL, and AbnQ recognized the importance and applicability of this contribution. We carefully addressed their suggestions as follows:

* Clarified the novelty of our approach, the design of the Token Decoder, the dataset generation process, potential limitations of mask usage, and the model's efficiency.

* Demonstrated generalization beyond synthetic benchmarks through additional evaluations on human-generated infeasible instructions, confirming CAMILA’s robust performance in real-world scenarios.

* Provided a comparison against Step1X-Edit, a recent state-of-the-art image editing model, on the Context-Aware Image Editing dataset, showing that CAMILA performs more effectively on context-aware tasks by achieving higher accuracy across all metrics while maintaining significantly lower runtime.

Overall, CAMILA shows superior performance compared to existing image editing models across multiple metrics, introduces new benchmarks for context-aware and multi-instruction editing, and offers a practical, efficient framework for tackling a key challenge in real-world image editing tasks.
We once again thank all reviewers and the area chair for their time and constructive feedback, which greatly helped us improve the quality of the paper.

Best regards,

The Authors

---

### Decision · Program_Chairs · 2025-09-17

**Decision:**

Accept (poster)

**Comment:**

The paper proposes a method CAMILA for the masked image modeling task. This method constructs a Context-Aware Aggregation module, which improves the reconstruction of masked tokens by leveraging both global context and local neighborhood interactions. Several reviewers agree that this method is simple but interesting and that it shows effectiveness on the evaluations. During the discussion phase, the authors provide further explanations and results, which address many reviewers' concerns. This paper is expected to reach the acceptance level of NeurIPS 2025, but authors need to complement the provided statements and experiments in the final version.